# Camera Trap Methods and Drone Thermal Surveillance Provide Reliable, Comparable Density Estimates of Large, Free-Ranging Ungulates

**DOI:** 10.3390/ani13111884

**Published:** 2023-06-05

**Authors:** Robert W. Baldwin, Jared T. Beaver, Max Messinger, Jeffrey Muday, Matt Windsor, Gregory D. Larsen, Miles R. Silman, T. Michael Anderson

**Affiliations:** 1Department of Biology, Wake Forest University, Winston-Salem, NC 27109, USA; rwbaldwin13@gmail.com (R.W.B.); messmc9@wfu.edu (M.M.); mudayja@wfu.edu (J.M.); larseng@wfu.edu (G.D.L.); silmanmr@wfu.edu (M.R.S.); anderstm@wfu.edu (T.M.A.); 2Wake Forest University Center for Energy, Environment, and Sustainability, Wake Forest University, Winston-Salem, NC 27109, USA; 3Department of Animal and Range Sciences, Montana State University, Bozeman, MT 59717, USA; 4Pilot Mountain State Park, North Carolina State Parks, 1792 Pilot Knob Park Rd, Pinnacle, NC 27043, USA; matt.windsor@ncparks.gov

**Keywords:** camera trapping, N-mixture modeling, density estimates, mark–resight, North Carolina, *Odocoileus virginianus*, population estimation, imperfect detection, drones, thermal imaging

## Abstract

**Simple Summary:**

Wildlife researchers and managers can choose from several techniques to estimate the number of animals in a population. Camera traps and drones are increasingly common and cost-effective options that can estimate the number and density of wildlife. We compared three simple methods that are used to estimate the number and density of deer in Pilot Mountain State Park: (1) mark–resighting and (2) N-mixture modeling, both using camera trap data, and (3) extrapolating transect counts from thermal videos collected from an aerial drone. We found that all three methods provided similar estimates of the population’s density, with complementary strengths and weaknesses: drone surveys collected data quickly and precisely, mark–resight ratios provided estimates of different demographic groups, and N-mixture modeling revealed changes in density across different habitat qualities and seasons.

**Abstract:**

Camera traps and drone surveys both leverage advancing technologies to study dynamic wildlife populations with little disturbance. Both techniques entail strengths and weaknesses, and common camera trap methods can be confounded by unrealistic assumptions and prerequisite conditions. We compared three methods to estimate the population density of white-tailed deer (*Odocoileus virgnianus*) in a section of Pilot Mountain State Park, NC, USA: (1) camera trapping using mark–resight ratios or (2) N-mixture modeling and (3) aerial thermal videography from a drone platform. All three methods yielded similar density estimates, suggesting that they converged on an accurate estimate. We also included environmental covariates in the N-mixture modeling to explore spatial habitat use, and we fit models for each season to understand temporal changes in population density. Deer occurred in greater densities on warmer, south-facing slopes in the autumn and winter and on cooler north-facing slopes and in areas with flatter terrain in the summer. Seasonal density estimates over two years suggested an annual cycle of higher densities in autumn and winter than in summer, indicating that the region may function as a refuge during the hunting season.

## 1. Introduction

Population surveys establish baseline information for a wide spectrum of ecological processes and traits, including population dynamics [1], species interactions [2], disease prevalence [3], and biodiversity [4]. Many methods can be used to survey and estimate populations by sampling or complete census; advancing technologies are adding and refining methods that observe wildlife with little or no disturbance, and advancing statistical techniques can relax past assumptions of modeling and parse the components of uncertainty into meaningful inferences [5,6]. However, both new and conventional methods must be validated in real conditions and against one another to ensure that they can provide accurate and appropriate inferences [7]. At large scales and under realistic management conditions, this validation can entail a high and potentially prohibitive cost and effort.

Remote photography surveys have surged in popularity since the development of affordable commercially available infrared-triggered camera traps [8,9,10]. Camera traps, once deployed, serve as long-term fixed observers operating continuously in a wide range of environmental and climatic conditions with minimal human attention [11]. Consequently, camera traps are less invasive, less labor-intensive, and more cost-effective than many other ecological monitoring techniques [10,12,13], and are often deployed *en masse* to survey large areas of habitat. Such large-scale surveys are sensitive to the number of cameras deployed, the density of their placement, and individual camera placement and perspective, as these contextual parameters can determine the effective sampling area of each camera trap and therefore the aggregate area and proportion of habitat surveyed. Additionally, data from camera traps can challenge efforts to estimate population size (the gross number of individuals in a population) or density (the number of individuals per unit of area) because camera traps do not achieve perfect or necessarily consistent sampling [14,15]. Because camera traps generally detect and photograph limited portions of their assumed sampling areas, resulting records conflate true absences of the target species with undetected occurrences [15,16]. Estimates that do not account for such ‘imperfect detection’ misrepresent target populations and can misinform management decisions. Several analytical methods account for imperfect detection of camera traps [10,17], but modified mark and recapture methods such as mark–resighting are most commonly used by managers. 

Mark–resighting is often used with camera trap data to estimate populations but is limited by the precondition that some individual animals be identifiable, such that the ratio of identified animals to all animals sighted can be used to estimate total abundance at each camera site [15]. Simple mark–resight methods operate under two assumptions: (1) that individuals are correctly identified and (2) that individuals have equal probabilities of detection [18]. The assumption of individual identification therefore restricts these methods to populations that have been previously marked for identification or species with individually distinct visual markings such as patterned pelage [19] or conspicuous antler patterns [20]. Many of these methods were chiefly characterized in the context of baited camera sites [20], which increase the probability of detecting local individuals but can problematically bias population estimates because movements and responses to bait can differ among demographic groups and change across seasons [21]. Modern mark–resight methods account for heterogenous detection of individuals, animal movement and spatially heterogenous habitat use, and imperfect detection [22], which improves the ability of mark–resight methods to accurately describe populations across space and time. However, many of these methods require fine-scale animal movement data whose collection entails higher costs to both researchers and animals, and many studies still estimate populations from simple mark–resight ratios instead [23].

N-mixture modeling (NMM) uses count data collected across time in multiple locations to estimate abundance without identifying individuals [24,25]. NMM treats the probability of photographically capturing individuals (i.e., ‘detection’) and true abundance independently by using variation in site-specific counts over time to estimate the probability that individuals are detected. NMM entails several assumptions: (1) that variation in point counts only results from the probability that individuals are detected [25]; (2) that populations are closed; (3) that there are no false-positive species identifications; (4) that detections are independent; and (5) that there is a constant, homogeneous detection probability for all individuals [16]. NMM can incorporate covariates to model both detection and occupancy, and it distinguishes non-detection from true absences [26]. In doing so, NMM can provide fine-scale variation in abundance to inform precise management actions. NMM has been used to estimate population characteristics of birds, amphibians, and tropical mammals [27,28,29], but its accuracy under field conditions remains contentious. 

Despite its popularity, NMM has been criticized as a method due to its dependence on unrealistic assumptions [30]. Repeat-counting of individuals across samples, population fluctuations within the survey period, and heterogenous detection probability among individuals or subclasses each violate assumptions of NMM yet are often unavoidable within field surveys [31]. Recent simulation studies show that these violations only marginally affect model fit but can radically change resulting population estimates [31,32]. Previous validations of NMM have relied upon simulated data that conform to the underlying assumptions of the model [25] and do not necessarily validate the model in field scenarios in which modeling assumptions may not be realistic or verifiable—for example, free-ranging populations of animals [33].

Recent decades have also yielded new drone-based methods for wildlife research proceeding from many of the same technological advancements that have enhanced camera-trap methods, including cheap high-resolution digital photography, improved thermal sensors, and on-board micro-computer systems for decision making and remote camera control. Drones can autonomously collect imagery along preprogrammed flight routes that sample or census large spatial extents of habitat. Depending on the drone’s payload, imagery can be collected as stills or continuous video; in color, multispectral, or thermal infrared spectra; and at a variety of resolutions (also depending on the camera and planned altitude of flight). Different payloads and imaging methods can entail trade-offs with respect to the data collected or can be combined to complement one another and provide novel information, and they are further contextualized by the choice of aircraft and flight routes—which can also determine the cost of the operation in money and effort and the potential disturbance to both target and non-target animals in the overflown area. In the case of large mammals, thermal imagery can exploit the difference between endotherms and their environments, particularly during twilight hours and winter months, to reveal animals that might otherwise appear cryptic in visible light [34]. Drawbacks of thermal imagery include higher cost sensors with lower resolutions compared to color cameras and potential ambiguity of the detected species if multiple species appear similar in thermal infrared radiation. Videography, in contrast to still photography, can additionally reveal animal movement, enhancing animal detection and classification against a comparatively inert environment. Tools and analytical methods for video-type data still lag behind those for still photography, but emerging techniques promise to integrate the strengths of continuous video with conventional photogrammetric methods [35]. When disturbance is a concern, fixed-wing aircraft generally produce a quieter acoustic profile than multirotor drones of similar size—though sometimes at the trade-off of a threatening visual profile (e.g., Egan et al. 2020 [36])—and flying at higher altitudes further decreases a drone’s acoustic and visual profile from the ground at the cost of a higher ground-sampling distance. 

The broad goals of this study were to compare wildlife survey methods with a free-ranging population of white-tailed deer (*Odocoileus virginianus*; hereafter deer), to derive density estimates for a moderately large (10.24 km^2^) extent of habitat, and to relate temporal fluctuations in regional density to ecological processes within a protected area. Deer are keystone herbivores in the eastern United States [37], and at elevated densities they can have profound homogenizing effects on understory composition and structure due to selective browsing pressure [38,39]. Given that male deer are individually identifiable by their antler characteristics [20], this species can therefore be used with mark–resighting, NMM, and aerial thermal videography to estimate population density using each method. Simple mark–resighting has been used widely to study deer [20,21], but NMM (commonly used for other species) has been sparsely applied to deer in a management context due to expected violations of the method’s assumptions [23]. However, white-tailed deer are a heavily managed game species, and their behaviors and use of space have been well characterized in a variety of environments [40]. As such, we designed a sampling scheme of unbaited camera traps to accommodate previously described occupancy characteristics of deer and reduce the probability of detecting the same individuals across multiple camera sites in an attempt to minimize violations of this assumption of NMM.

Our third method of population estimation—aerial thermal videography from a drone—provided a complementary method of estimating population density and a potential validation of mark–resight and NMM estimates from camera trap data. This implementation of drone surveillance has previously been used to estimate population density of white-tailed deer in a known closed population and otherwise natural setting [34], demonstrating the efficacy and accuracy of the technique for this species. Our study had two specific aims: (1) to compare camera survey estimates of populations (expressed as deer density) with one another and sampling from aerial thermal videography, and (2) to use NMM to relate fluctuations in deer density to spatial and temporal ecological processes at a local and regional scales. We hypothesized that each of the three methods would produce an approximately accurate estimate of the population based on previous successful applications of each method with populations of white-tailed deer. However, given the simplicity and coarseness of the mark–resight method, the controversial assumptions of NMM, and the novelty of counting from aerial thermal videography, we did not necessarily expect close agreement among the estimates. Past studies have varied with respect to their use of baiting [41], season [42], and closed or open populations [43], so it is difficult to compare among them. This study, therefore, addressed the need to test and validate these methods in a consistent, realistic management scenario.

## 2. Materials and Methods

We conducted camera trapping and drone flights on a mountain section of Pilot Mountain State Park (PMSP), NC, USA, 36.3425° N, 80.4768° W (Figure 1) during 2016–2018. The mountain section of the park consists of a quartzite monadnock that rises 738 m above sea level and ~425 m above the surrounding countryside. Differing levels of soil moisture and exposure among Pilot Mountain’s south/west and north/east slopes drive vegetational differences along these aspects. Pine-oak/heath dominate on the south/west-facing slopes as a result of xeric conditions from prevailing southwestern winds and greater afternoon sun exposure. Dominant species include Table Mountain pine (*Pinus pungens*, Lamb.) and pitch pine (*P. rigida* Mill.), with an understory of mountain laurel (*Kalmia latifolia* L.) and purple rhododendron (*Rhododendron catawbiense* Michx). Scrub oak (*Quercus ilicifolia* Wangenh.), blackjack oak (*Q. marilandica* Muenchh.), and chestnut oak (*Q. montana* Willd.) are also present, with an herb layer comprising mainly beetleweed (*Galax urceolata* (Poir.) Brummitt) and trailing arbutus (also known as mayflower; *Epigaea repens* L.). On cooler, more mesic east- and north-facing slopes the dominance shifts from pines to oaks (predominantly *Q. montana*), with a poorly developed herbaceous layer of primarily beetleweed [44]. Nearly all areas of PMSP are accessible to visitors, but most human activity is concentrated at a visitor’s center at the eastern boundary, the central peak, and a trail that links the two sites.

We established a grid of camera trap sites (*n* = 22) across a 10.24 km^2^ (2530 acre) study region of PMSP (Figure 1). We separated camera sites by approximately 636 m to resemble the common management practice of 1 camera per 100 acres [45,46]. Assuming that individual deer occupy a core range of 0.24 km^2^ for >50% of their time [47], this camera density of ~2.4 cameras/km^2^ established a low probability of capturing individuals at multiple camera sites while achieving adequate coverage of the study area to feasibly detect all uniquely identifiable individuals within the park. We chose original site locations using a systematic design on a randomly generated grid and made minor *ad hoc* adjustments to place cameras at sites with low immediate steepness and easy access by foot [20]. We used Cuddeback E3 model cameras (Non Typical, Inc., Green Bay, WI, USA) set with “high” trigger sensitivity and “optimal” detection range. We established camera sites in June 2016 and maintained them continuously until September 2018 by downloading data from memory cards and changing batteries regularly to achieve continuous monitoring. We mounted cameras 60–90 cm above the ground and facing north to reduce backlighting at sunrise and sunset, we cleared obstructive vegetation from each camera’s view to reduce the likelihood of detected movement from wind gusts, and we mounted plastic identification tags opposite each camera to facilitate site recognition from imagery. All sites recorded deer during the deployment period, and humans and coyotes were also recorded on both camera traps and aerial videography. All species were visually distinct in camera trap imagery; humans were visually distinct from deer in aerial thermal videography, and coyotes were sufficiently rare in camera trap data that they were considered a trivial source of potential error in counts from aerial thermal videography. 

For mark–resight analysis we counted the number of adult males, adult females, fawns, and individuals of indeterminable demographic characteristics in each photograph between 1 December 2017 and 28 February 2018—a period that corresponded to the winter bin of our NMM seasonal analysis. We then applied the method described by Hamrick et al. [45] to estimate the density according to a common method among wildlife management practitioners. We identified all unique male deer photographed across all sites during this period and calculated a “population factor” as the ratio of identifiably unique male individuals (identified by antler characteristics) to all occurrences of adult males (antlered adults) in photographs.
population factor = unique adult males ÷ all adult male occurrences(1)

We then multiplied this population factor by all occurrences of presumed adult females (antlerless adults) in photographs to estimate the number of unique females among camera-trap photographs.
unique females = population factor × all adult female occurrences(2)

Finally, we multiplied the population factor by all occurrences of fawns in photographs to estimate the number of unique fawns among camera-trap photographs.
unique fawns = population factor × all fawn occurrences(3)

We estimated the population of our sampled area by summing these estimates and applying an extrapolation factor to correct for undetected individuals in our population.
population estimate = 1.10 × (unique males + unique females + unique fawns)(4)

Based on the 3-month duration of our observation period, we used a conservative extrapolation factor of 1.10 (which has been empirically estimated for baited surveys of this length [45,48]) in the absence of a recommended extrapolation factor for unbaited surveys—which, we expected, were less likely to attract individuals in front of the camera and might therefore warrant a larger extrapolation factor. We then calculated the population density by dividing this total estimate by the study area (10.24 km^2^) [46].

We generated 95% confidence intervals for mark–resight estimates from a bootstrapped distribution of 10,000 density estimates calculated using bootstrapping: we randomly resampled 10,000 combinations of our sites (*n* = 22) with replacement and then aggregated the resulting mark–resight abundance estimates of those sites into a total density estimate across sites for the bootstrap instance. The resulting distribution of resampled density estimates yielded an estimated confidence interval of the total density estimate. 

For each day and each site, we counted deer in each photograph and used the highest count to represent that day and site in NMM in order to avoid the risk of counting the same individuals multiple times and to disregard photographs in which the frontmost animals obstructed the camera’s field of view. We binned observations into three-month seasons—summer (June–August), autumn (September–November), winter (December–February), and spring (March–May)—to model temporal changes in deer abundance and to control for seasonal differences in animal behavior and the related probability of detection. We used 30 m resolution elevation data from the USGS National Elevation Dataset [49] to calculate the site-specific elevation, slope, aspect, and distance to the nearest park edge for each camera site. We transformed the aspect from an angular variable (0–360°) into its north–south component by taking the cosine of the angle, which we hereafter describe as the aspect.

We used NMM with a hierarchical form:*N_i_* ~ *Poisson*(*λ_i_*); with log(*λ_i_*) = *β*_0_ + *β*_1_*C_ij_*|*N_i_* ~ *Binomial*(*N_i_*,*p_ij_*); with log(*p_ij_*) = *α*_0_ + *α*_1_(5)
where *N* is the latent abundance of site *i*, *C* is the point count at site *i* and time *j*, *λ* is the mean abundance, and *p* is the detection probability of individuals at site *i* and time *j*. Covariates (*β*, *α*) were incorporated into the detection and abundance portions of the model using log-link functions [25].

We created candidate models that included one or two environmental predictors (elevation, slope, aspect, and distance to edge) independently in both the detection and abundance components of the model (such that all 42 possible combinations were represented as candidates) as well as a null model with no covariates included (Appendix A). We selected a single model for each three-month season from summer 2016 through summer 2018 using Aikake’s information criterion (AIC). For each site, we calculated an empirical Bayes best unbiased predictor of mean abundance from a simulated posterior distribution of the top performing model along with 95% credible intervals. We estimated the population density of deer in our study area using NMM each season by adding the mean abundance estimates of all camera trap sites and dividing the sum of estimates by the total area sampled across all cameras (10.24 km^2^). We also modeled deer abundance from camera trap collected between 15 January and 15 March 2018—a period approximately centered around the timing of our drone surveys—using the selected model structure for winter 2017/2018 to assess agreement between the NMM estimates and the estimate from aerial thermal videography.

We obtained normalized difference vegetation index (NDVI) values from NASA’s Moderate Resolution Imaging Spectroradiometer (MODIS) at a spatial resolution of 250 m and temporal resolution of 16 d (product MOD13Q1 v006) for each camera site and averaged values by season to estimate changes in vegetation greenery over time [50]. We analyzed all data using R version 3.4.0 [51] with the packages *unmarked* [52] for NMM, *AICcmodavg* [53] for model selection, *raster* [54] to generate topographic products, and *MODIS* [55] to generate products of vegetation dynamics. We report model estimates with standard error (SE) and *p*-values with an α = 0.05.

We flew drone surveys in February 2018 to obtain a third estimate of population size and density through direct counts. We flew five replicate surveys using a Linn Aerospace Hummingbird quadcopter equipped with a non-radiometric thermal infrared imager with resolution of 640 × 512 pixels (FLIR Vue Pro 640; 13-mm lens, 45° horizontal field of view, 30 Hz video framerate). These surveys took place on 8 February, AM; 8 February, PM; 9 February, AM; 9 February, PM; and 13 February, AM. All flights occurred within one hour of sunrise or sunset. We selected these dates to maximize the detectability of animals during a period when temperate trees were devoid of leaves and deer body temperatures contrasted against background temperatures in thermal imagery [34]. We flew all surveys at 120 m above ground level and achieved a 100-m horizontal field of view (15.6 cm/pixel ground sample distance). The aircraft flew a long corridor of 12 parallel transects oriented along an east–west axis spaced evenly 300 m apart across the study area to avoid possible double counts. The start location of the flight was randomized to achieve a systemically randomized design of parallel, equally spaced transects throughout the study region. Transects varied by length, and northern transects were aborted when the aircraft lost line-of-sight communications with its controller (Figure 1). These transects summed to 13.75 km in length and surveyed 1.375 km^2^ or 13.5% of the study area. During each flight, the drone continuously collected thermal video, which we processed and analyzed according to the methods of Beaver et al. [34]. The thermal video showed uneven temperature measurements across the field of view resulting from uneven cooling of the sensor during flight; however, this did not appear to interfere with visual identification of deer’s thermal signatures during video analysis. Using the resulting counts, we first calculated the mean density of deer per transect in each flight; using the summed transect densities, we calculated the mean density of deer in each flight; and using the flight densities, we calculated the mean density of deer in our study region across all flights. We then bootstrapped the mean densities from each flight (*n* = 5) to generate a 95% confidence interval of our density estimate from aerial thermal videography. We used an analysis of variance (ANOVA) using a type II sum of squares to test for significant differences between flight estimates.

## 3. Results

The camera traps collected 16,775 photographs of deer over our total 27-month study period (after false-positive photographs were removed) and 1695 photographs of deer specifically during the three-month period from 1 December 2017 to 28 February 2018. The mark–resight analysis and NMM using camera trap data and the aerial thermal videography each produced density estimates that were comparable (Figure 2). The aerial thermal videography estimated 31.3 deer/km^2^ (95% CI: 23.7–39.0) in February with no significant differences between flights (F4,28 = 1.09, *p* = 0.30). The NMM estimated the 27.5 deer/km^2^ (95% CI: 19.9–36.9) during the overlapping period (15 January–15 March 2018) and 40.7 deer/km^2^ (95% CI: 31.9–51.0) during winter 2017/2018; the mark–resight analysis estimated 27.7 deer/km^2^ (95% CI: 15.4–48.7) for the same period of winter 2017/2018. We determined that the NMM and drone estimates for the month of February were statistically indistinct because both means occurred within the 95% confidence interval or credible interval of each another (Figure 2).

Selected seasonal models from NMM contained different combinations of predictors in both the abundance and detection components of the model, but key patterns emerged. Slope was the most common predictor of abundance among the selected models, most often with an inverse relationship (higher abundance on shallower slopes); the elevation and aspect were also commonly selected predictors (Table 1). Coefficients associated with aspect were negative in the winter (suggesting a preference for drier, pine-dominated, south-facing slopes) and positive in the summer (suggesting a preference for cooler, oak-dominated, north-facing slopes). For the winter 2017/18 season, when all three density estimation methods were compared, higher deer densities occurred at sites characterized by a southerly aspect and high elevations (*β*_aspect_ = −0.26, SE = 0.07, *p* = 0.0004; *β*_elevation_ = 0.23, SE = 0.09, *p* = 0.01). Elevation was the most common predictor of detection, most often with a direct relationship (greater detection at high elevations). Slope was the next most common predictor of detection with a consistently inverse relationship (greater detection on flat slopes). In the winter 2017/18 season, slope was a significant predictor of detection with an inverse relationship (greater detection on flat slopes; *α*_slope_ = −0.29, SE = 0.09, *p* = 0.001).

Our temporal analysis using NMM revealed regular seasonal fluctuations in regional deer density over the course of our 27-month study period. PMSP’s deer density was lowest in the summer months, increased steadily in the autumn and winter months, and slowly decreased throughout the spring—varying inversely with seasonal NDVI patterns (Figure 3). The amplitude of change differed across years, but the general trend appeared in both years, and density estimates in winter months approximated those of aerial thermal videography in February 2018.

## 4. Discussion

All three methods of estimating population density produced similar estimates—two using camera trap datasets and one using aerial thermal videography. This corroboration among estimates suggests that all three methods converged on an accurate representation of the population, although replicate surveys would add additional confidence to this interpretation. The small differences between estimates were likely due to the various assumptions entailed with each method. Mark–resight estimates of deer density require the photographic capture of all individual male animals and account for imperfect detection by inflating estimates by an extrapolation factor [45] as informed by baited surveys of captive populations [20,48]. Notably, although a variety of mark–resight methods have been proposed, the calculations used in this study were representative of common practices in deer management. The simple ratios were applied to raw counts with empirically estimated extrapolation factors [45] rather than estimated encounter rates [20] and did not account for potential variance in encounter rates across sites [56], between sexes [57], or across seasons [58]. Furthermore, proper extrapolation factors have not been explored for unbaited mark–resight surveys, which likely capture fewer animals due to the absence of an attractant. Our use of a conservative extrapolation factor (one that expects more complete surveillance) likely yielded lower density estimates than other methods and the relatively large difference from the NMM estimate of the same time period (Figure 2). NMM accounts for uncounted individuals by estimating imperfect detection probability and abundance separately [25], yielding more accurate density estimates if the method’s assumptions are upheld. NMM has been validated on large, free-ranging ungulates using telemetry data to estimate immigration and emigration rates [59], but individual movement data are time-intensive, costly, invasive, and rarely available to stakeholders. Our findings applied NMM using a freely available, open-source software [52] to achieve validated abundance estimates for free-ranging ungulates and to test hypotheses regarding abundance relationships with spatial and temporal predictors. Ultimately, aerial thermal videography provided density estimates with the highest confidence informed by prior validation of the method [34], a relative consistency among replicate surveys, and a comparative lack of assumptions necessary to infer abundance from video-derived counts over sampled transects. Based on the visual appearance and contrast in the thermal video data that we collected, we presumed that analysts detected all deer that occurred along each transect strip. This methodology could be improved, however—especially when there is poor contrast among thermal signatures—by incorporating distance estimates with deer counts across the camera’s field of view; this would enable analysts to identify and correct for any decrease in detection across distance from the center of the camera’s perspective.

The seasonal influence of aspect on deer abundance highlighted the important influence of spatial context on population surveys. Aspect can determine soil content of nutrients and organic matter due to directional exposure to wind and sunlight and moisture regimes [60], which can produce contrasting compositions of the vegetation community between different aspects [61]. Deer preferentially select areas of higher relative nutritive quality when such habitat is available among other areas of equivalent cost and lower quality [62]. Additionally, white-tailed deer adjust their behavior and habitat use to avoid thermal stress and balance foraging needs against the risk of heat loss during colder periods [63,64]. Within our study area, individuals occupied southern slopes during winter months, likely due to their higher amounts of available browse within the mixed forest overstory area [65] and their warmer temperatures. In highly seasonal environments, deer select habitats by aspect during winter months, but this has been attributed to lower accumulation of snow on slopes that receive more sunlight [66,67]. In our study system, southern slopes additionally provide easier access to nearby agricultural lands, whereas northern slopes are bounded by suburban areas. Habitat preferences of deer are already well documented, but our results demonstrated that NMM can further reveal spatial influences on abundance in a robust manner and over multiple seasons. 

We propose that the seasonal fluctuations in deer density in PMSP (Figure 3) describe changes in habitat use and selection rather than local changes in detectability. The range and core habitats of white-tailed deer often fluctuate throughout the annual cycle, but the population of PMSP is known to be non-migratory, and such shifts are expected to be relatively minor in this habitat [40,68,69]. Rather, the seasonal fluctuation that we observed may reflect the interplay between risk and resource availability in habitat selection at a local scale. Ungulates experience a trade-off between mortality risk and forage quality when they select habitat [70,71], prioritizing safety over forage when risks of predation are especially high [72]. In the resource-rich agricultural landscapes that surround our study area, recreational hunting pressure poses a substantial risk to deer during the autumn hunting season, which deer perceive and to which they respond [73]. Public lands in the western United States shelter abundances of ungulates disproportionate to the regions’ areas during hunting seasons due to the restriction and prohibition of hunting on those lands [74,75]. Our temporal analysis suggests that PMSP may function as a similar refuge for local deer. Conversely, during the early growing season, the deer density of PMSP gradually decreases as habitat preference shifts into surrounding lands with less tree canopy cover and greater forage availability at ground level. These seasonal trends in deer density demonstrate the importance of considering behavior and landscape context when interpreting population surveys; survey methods and designs should account for these factors alongside the management goals of the study [15]. Longer multiyear studies and individual-scale movement tracking can clarify the degree to which seasonal shifts in habitat selection manifest in local density estimates.

High temporal and spatial coverage are necessary for NMM but can present tradeoffs to a study’s cost and effort [76]. A high range and density of camera coverage facilitates the detection of rare or cryptic species, whereas a long period of continuous or targeted monitoring allows for precise estimates of common species [24]. The consistency of our density estimates across estimation methods suggests that we achieved satisfactory coverage of our study region using a camera placement schema based on existing knowledge of spatial habitat use by our focal species—home ranges and core area sizes of deer are extensively detailed in published literature [40]. In Appalachian forests, deer occupy an average core area of use of 0.24 km^2^ and a home range size of 10 km^2^ during winter [47], but home ranges and space use are also known to vary with habitat quality [77], season [78], and sex [64]. Among these sources of variation, we selected a fixed camera density (2.4 camera/km^2^) while attempting to balance goals of (1) a low probability of detecting individuals across multiple camera sites in our study area for NMM analysis with (2) achieving adequate coverage to capture a high number of individually identifiable animals for mark–resight analysis. A mark–resight analysis of uniquely identifiable individuals could inform estimated rates of repeat-counting in male deer among photographs used for NMM and reveal potential changes across seasons; however, such an analysis would demand considerable effort from analysts to catalog and document all identifiable males across all photographs based on their antler patterns each year, which was not feasible within the scope of this work. Our decision to only analyze the highest count of deer photographed at each site in each 24-hour period methodologically did reduce the risk of repeat-counting individuals within the daily sample period, which aligned with the diel activity patterns of deer [64]. However, this data selection practice would not be appropriate for species and populations that form herds on a seasonal basis, as larger aggregations might be misconstrued to reflect a change in abundance rather than behavior. Such changes in social behavior might manifest in counts of group sizes across all photographs in the effort, but such changes were not expected, noticed, or explicitly tested in this work.

We cannot ensure complete closure between our camera sites, and we did not calibrate or validate their true effective sampling area, so our interpretation of abundance changes is not an estimate of absolute abundance but is rather most appropriately interpreted as an estimate of habitat use [79]. Site-specific abundances for each camera location represent the mean number of animals that used that local site in our study area and not necessarily the “true” population density of that site and overlapping population. The relative congruency among our three estimation methods suggests that repeat-counting of individuals either did not occur often or did not affect our estimates using NMM if it did. Notably, this application of NMM was sensitive to both the density of camera trap placement and the bins used for temporal aggregation, and future applications of this method should consider the behavior and ecology of the target species and should test the sensitivity of these parameters in the study’s design. Extensions of NMM can address heterogeneity within sample units, alternative Poisson mixed model structures can relax the assumptions of the estimation, and occupancy modeling methods can yield useful occurrence proxies under imperfect detection conditions [30,32]. Even when acknowledging the limits of our chosen methods, congruency across the resulting density estimates suggests that a basic NMM implementation, simple mark–resight ratios, and a targeted survey of aerial thermal imagery constitute low-effort methods of estimating a deer population over a moderately large (10.24 km^2^) region of habitat and—in the case of NMM—can yield limited spatial inference to inform integrated landscape management.

The simplest, most cost-effective, and likely most accurate methods, however, are simple counts from drone-based thermal videography, which in this study captured population density within a few hours of flights—with subsequent processing and analysis—and has been shown in studies of known captive animals to reproduce known population sizes with high fidelity [34,80,81]. Flights repeated seasonally could capture many of the spatiotemporal patterns revealed by long-term camera trapping and NMM analysis, and the spatial quality of aerial videography enables georeferenced observations of animal movement and short-term patterns of habitat use [35]. A major drawback of aerial thermal videography, however, is its inability to describe other aspects of deer populations such as sex and age or serendipitous encounters with non-target animals, which can reveal additional ecological insights.

## 5. Conclusions

The goals of management action can determine the most suitable survey methods for an applied scenario. All three methods employed in this study are practical options for species that exhibit human avoidance or occupy habitats that cannot be surveyed directly by ground. Mark–resight surveys can leverage short survey windows and flexible study designs to achieve simple estimation of demographic information such as age- and sex-structures in a population compared to NMM and other modeling methods. Such estimates can critically inform decisions for population management, which often target demographic classes and processes [82]. However, it can be difficult to calculate the precision of mark–resight estimates, requiring replicate surveys that are often impractical or unachievable. In the example of our study, the error associated with mark–resight estimations exceeded that associated with each other method (Figure 2). Managers must be able to compare estimations across space and time to evaluate ongoing management actions [83], and NMM can estimate the precision of its estimates, thereby enabling such comparisons among estimates. A combination of approaches might be appropriate to achieve the management goals and resources available in a given area, and our findings suggest that in a moderately large study area and with species-tailored sampling schemes, managers have multiple surveillance techniques at their disposal to achieve consistent density estimates, including camera-trap sampling and aerial imaging from drones.

## Figures and Tables

**Figure 1 animals-13-01884-f001:**
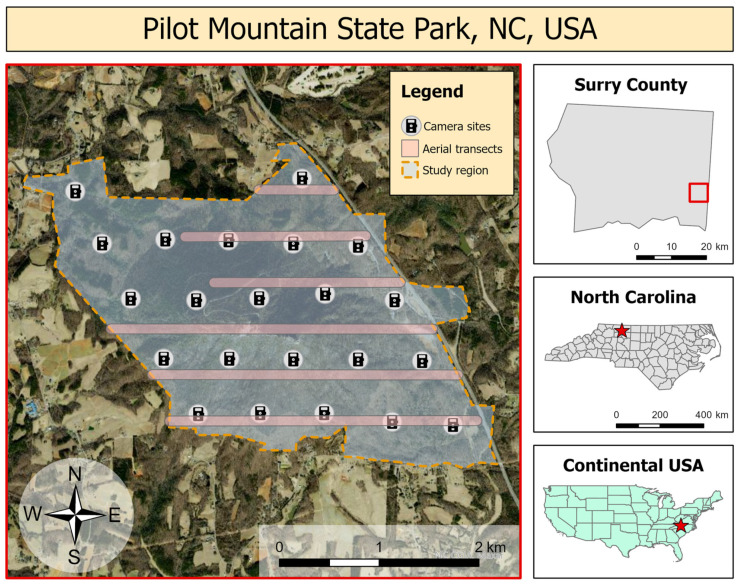
The study area in PMSP, North Carolina, USA. Camera sites were spaced approximately 636 m from one another for a density of 2.4 cameras/km^2^ over a total area of 10.24 km^2^. Each site was outfitted with a single camera facing north that collected data continuously between June 2016 and September 2018. Drones collected continuous thermal video along aerial transects from an altitude of 120 m and recorded a swath of 100 m in 15.6 cm/pixel ground sample distance for a total surveyed area of ~1.375 km^2^. Insets indicate the location of the study area (red stars) within North Carolina and the continental USA.

**Figure 2 animals-13-01884-f002:**
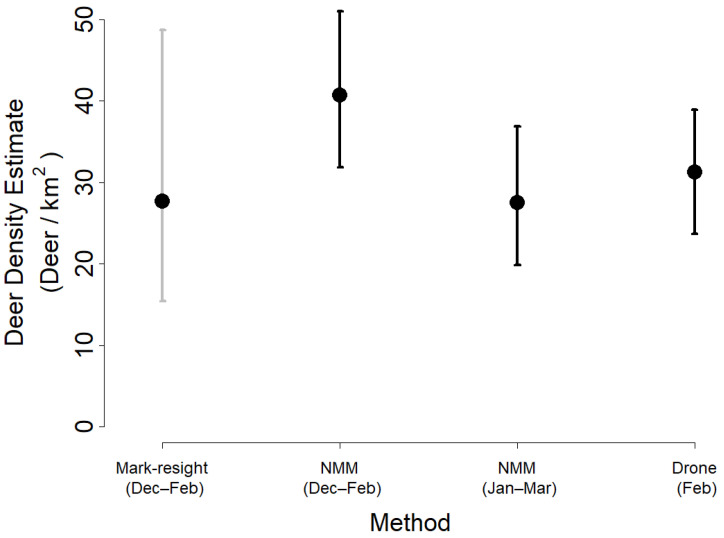
Mean population density of deer as estimated by mark–resight, NMM, and aerial thermal videography (drone) methods during surveys of PMSP in 2017–2018. Mark–resight methods and NMM were used to analyze camera trap photographs collected during 1 December 2017–28 February 2018; NMM was also used to analyze camera trap photographs collected during 15 January–15 March 2018; and aerial thermal videography was collected during 8–13 February 2018. All error bars represent 95% confidence intervals or credible intervals.

**Figure 3 animals-13-01884-f003:**
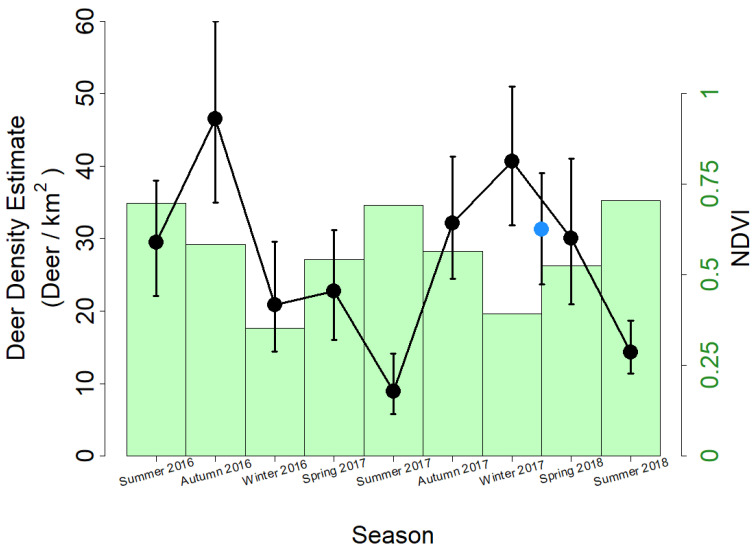
Seasonal changes in the deer population and vegetation in PMSP. Deer density was estimated using NMM with camera trap data (black points) and aerial thermal videography (blue point); connecting lines illustrate trends over time. Vegetation intensity was estimated using NDVI values (green bars) from MODIS satellite imagery [50]. Three-month seasonal bins were used to aggregate the camera trap data for NMM and to calculate average NDVI values. Error bars represent 95% credible intervals of the mean best unbiased predictor of site abundance for estimates using NMM or a 95% confidence interval bootstrapped from mean density estimates of replicate surveys with aerial thermal videography (*n* = 5).

**Table 1 animals-13-01884-t001:** Parameters estimates, SE, and associated statistics of selected N-mixture models that estimated the density of deer in our study region of PSMP, North Carolina, for each season between June 2016 and August 2018. Maximum counts were determined from all photographs in a 24-h survey windows each day for each camera trap site (*n* = 22) and binned into three-month seasons. Models were then fit to these seasonal count subsets and predictors were selected using AIC (Figure 1). *B*_-Abundance_ and *B*_-Detection_ represent intercepts for the abundance and detection components of each N-mixture model.

Season	Parameter	Estimate	SE	Z-Value	*p*-Value
Summer 2016	*B*-_Abundance_	2.19	0.17	12.99	<0.001
	Slope	−0.32	0.13	−2.46	0.014
	*B*-_Detection_	−3.12	0.17	−18.68	<0.001
	Edge	0.4	0.2	2.04	0.042
	Elevation	−0.43	0.14	−3.08	0.002
Autumn 2016	*B*-_Abundance_	2.68	0.16	16.74	<0.001
	Elevation	−0.25	0.1	−2.55	0.011
	Slope	0.19	0.1	1.99	0.046
	*B*-_Detection_	−3.73	0.16	−23.48	<0.001
	Aspect	0.17	0.07	2.49	0.013
Winter 2016/17	*B*-_Abundance_	1.87	0.15	12.12	<0.001
	Aspect	−0.2	0.1	−2.03	0.042
	*B*-_Detection_	−3.65	0.14	−25.4	<0.001
	Elevation	0.6	0.18	3.28	0.001
	Slope	−0.47	0.16	−2.91	0.004
Spring 2017	*B*-_Abundance_	1.94	0.16	11.78	<0.001
	Edge	−0.29	0.12	−2.33	0.02
	*B*-_Detection_	−3.34	0.16	−21.2	<0.001
	Elevation	1.23	0.23	5.45	<0.001
	Slope	−0.99	0.19	−5.3	<0.001
Summer 2017	*B*-_Abundance_	0.99	0.21	4.75	<0.001
	Aspect	0.36	0.16	2.2	0.028
	*B*-_Detection_	−3.39	0.19	−18.31	<0.001
	Elevation	1.08	0.21	5.06	<0.001
	Slope	−0.29	0.18	−1.57	0.115
Autumn 2017	*B*-_Abundance_	2.22	0.16	13.82	<0.001
	Edge	−0.22	0.12	−1.89	0.059
	Slope	−0.34	0.13	−2.58	0.01
	*B*-_Detection_	−3.21	0.15	−20.7	<0.001
	Aspect	−0.2	0.08	−2.56	0.01
	Elevation	0.3	0.14	2.05	0.04
Winter 2017/18	*B*-_Abundance_	2.51	0.14	18.54	<0.001
	Aspect	−0.26	0.07	−3.52	<0.001
	Elevation	0.23	0.09	2.49	0.013
	*B*-_Detection_	−3.19	0.13	−24.53	<0.001
	Slope	−0.29	0.09	−3.3	<0.001
Spring 2018	*B*-_Abundance_	2.22	0.2	11.01	<0.001
	Slope	−0.29	0.13	−2.32	0.02
	*B*-_Detection_	−3.76	0.2	−18.84	<0.001
	Elevation	0.4	0.13	2.98	0.003
Summer 2018	*B*-_Abundance_	1.56	0.15	10.21	<0.001
	Elevation	0.15	0.12	1.26	0.208
	*B*-_Detection_	−2.71	0.13	−20.76	<0.001
	Aspect	−0.3	0.11	−2.83	0.005

## Data Availability

Data and fully reproducible models are available via Github at: https://github.com/gl7176/PMSP_deer_survey_2016, accessed on 19 April 2023.

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
