# Peer review of "Camera Trap Methods and Drone Thermal Surveillance Provide Reliable, Comparable Density Estimates of Large, Free-Ranging Ungulates"

_animals, 2023, doi:10.3390/ani13111884_

Round 1
Reviewer 1 Report
Very well-written paper on emerging important techniques for wildlife scientists.
Perhaps define/differentiate upfront on the difference between deer density and deer numbers.
I am intrigued by the idea of deer using protected areas as refuges from hunting. I have observed that deer overpopulation issues typically occur in areas where they are not hunted (parks, suburbs, protected areas).
Author Response
We thank the reviewer for their favorable reception of our work, and for their helpful suggestions.
We have added text to clarify the difference between population size and density at the first instance of use (Line 63-64).
Reviewer 2 Report
The MS is a useful piece of scientific literature, and it deserves to be published. I have a number of revisions to ask before final acceptance.
1. Lines 42-49. Add some references.
2. Lines 154-165. This part lacks predictions, which may put your MS in a more hypothesis-driven context.
3. Lines 167-184. Plant names require the name of the descriptor. Please correct this part.
4. Figure 1 is unclear, please add a total USA map to better localise North Carolina.
5. Lines 315-317. How many other species did you record? In how many sites did you record deer?
6. Throughout the text and tables, change “fall” with “autumn”.
7. Lines 370-372. Please tone down this sentence. Similar estimates might also be all wrong.
8. In the conclusion, consider your methods to count wildlife where human-wildlife conflicts arise, or in areas where direct counts are challenging, e.g. mountains (see for instance Bocci A., Lovari S., Khan M.Z., Mori E. (2017). Sympatric snow leopard and Tibetan wolves: coexistence of large carnivores with human-driven potential competition. European Journal of Wildlife Research 63: 92).
Author Response
We thank the reviewer for their favorable reception of our work, and for their helpful advice regarding revisions. We have revised the text throughout, adding many more references, rewriting sections of the methods, and reframing our conclusions. We believe that these revisions have significantly improved our manuscript, and we thank the reviewers for their guidance.
We have specifically attempted to address each point from this reviewer with the following revisions:
- Lines 42-49. Add some references.
We have added three references that support these statements (Lines 47, 49)
- Lines 154-165. This part lacks predictions, which may put your MS in a more hypothesis-driven context.
We have added a hypothesis and prediction to the introduction (Line 170-179)
- Lines 167-184. Plant names require the name of the descriptor. Please correct this part.
We have added the descriptor for each plant name (Lines 190-194)
- Figure 1 is unclear, please add a total USA map to better localise North Carolina.
We have added a map of the USA to the figure (Line 201)
- Lines 315-317. How many other species did you record? In how many sites did you record deer?
We have noted the rare occurrence of humans and coyotes in camera trap and drone data, and that all sites recorded deer (Lines 229-234)
- Throughout the text and tables, change “fall” with “autumn”.
We have implemented this change throughout
- Lines 370-372. Please tone down this sentence. Similar estimates might also be all wrong.
We have toned down the statement, adding qualifications and limiting our interpretation of the results (Lines 419-424)
- In the conclusion, consider your methods to count wildlife where human-wildlife conflicts arise, or in areas where direct counts are challenging, e.g. mountains (see for instance Bocci A., Lovari S., Khan M.Z., Mori E. (2017). Sympatric snow leopard and Tibetan wolves: coexistence of large carnivores with human-driven potential competition. European Journal of Wildlife Research 63: 92).
We have added a sentence to this effect (Lines 564-566)
Reviewer 3 Report
General comments:
I agree with authors in that too many proposed methods to estimate population size are based on simulations and need verification in the field. Without having access to a known marked population comparing various method against each other appears to be a useful way forward. The three methods employed here are relatively “simple” to implement and the authors have done a decent job in listing the caveats and assumptions they entail. There are a few details I would have liked to be added (see below) and I think the analysis of the drone survey could be improved by accounting for imperfect detections via distance sampling.
Specific comments:
L186 & L195 I appreciate the nominal distance between cameras provided, but I doubt the accuracy down to 20cm. Moreover, camera placement in the field would have changed somewhat from the predetermined location (L202). Therefor the distance between cameras should also be variable and the authors could provide an average plus error term or perhaps a range.
L189 It should be 0.32
L196 I cannot quite follow the argument here. I understand that the authors used half the nominal distance between cameras the calculate a circle around each camera as a sampling area. I am not quite clear on the idea that the monitoring area for the cameras are touching but not overlapping, that the monitoring area is circular and that there are gaps between these circles as they don’t overlap. In other words, the determination of an overall survey area simply based on a somewhat arbitrary distance between cameras is opaque to me. However, I do understand that the size of the survey area is quite important to determine deer density and it would be good if the authors could elaborate in this crucial step. Furthermore, did the authors verify that none of the identifiable deer were not seen on more than one camera?
L218 onwards I get the general idea, but the description of the method in the text could do with overhaul – I got lost. Besides, I gather that for this estimate to work, detection probabilities for males, females, and fawns have to be the same. Given that the home range size of males is about twice as large as that of females that assumption appears somewhat questionable. The literature suggests that dependent on camera spacing either males or females can be favoured. I would like to see some discussion of this point.
L229 What does this extrapolation factor of 1.1 account for and is it really applicable to this study? Should it be used given the quite different scenario it was derived from (see discussion)?
L231 onwards The bootstrap approach could do with a bit more detail too. What variable was actually sampled, daily counts per site?
L240 I understand that using the highest snapshot count per day avoids double counting individuals, but in that case the population estimate does rely on animals aggregating into herds. I would assume that the propensity of males and females to form herds follows a yearly cycle. Hence, could the yearly cycle in density be caused not only by more animals being present, but deer aggregating in larger or smaller groups? The reliance on a maximum group size rather than a count of individuals appears somewhat problematic and the authors should discuss this issue somewhere in the manuscript. At the very least an annual cycle in group size for either sex should be explored.
L296 Given the method used, the authors assumed that all deer were detected along the transect strip. While that might be correct, the authors should check for a decrease in detections with perpendicular distance from the drone (i.e. distance sampling). Should there be a fall-off resulting in a non-uniform detection function across the field of view of the camera than this bias has to be corrected for in, for example, the software package ‘Distance’ or the equivalent R-package.
L349 There seems to something wrong with the table regarding Fall 2017. I am also confused about “entry 4”?
L378 I can see the conundrum of the correction factor and given the uncertainty about its application for the present setup, I probably wouldn’t have used any. However, the authors have the advantage to have monitored the deer over several years. I know it is a lot of work, but would it be possible to compare the detection of the various identifiable male deer over different time periods, going beyond three months the get an idea about the persistence of individuals in the data set, detection rate and therefore estimate how likely it was to have missed an individual in a three-months timeframe?
L390 Indeed aerial surveys have long been used to estimate the abundance and density of large herbivores. Usually however, detection along the survey strip is imperfect and has to be corrected for (distance sampling). Furthermore, transect placement should also be randomised to a degree to ensure that the sampled areas are representative of and can be extrapolated to the whole study area. It is not implausible that the narrow strip counts at a relatively high altitude yield perfect detections, but that should have been verified.
L410 As NMM in this instance relies on group size to a degree, I am not sure whether this conclusion is entirely justified. Annual migration into the reserves is certainly plausible, but annual variations in model performance might be a contributing factor. Perhaps the authors could include a short discussion to this effect.
L433 It was 0.25 in the introduction.
L434 What about differences between the sexes?
L436 Was the low probability confirmed during the study with none of the identified deer appearing on more than one camera?
Author Response
We thank the reviewer for their favorable reception of our work, and for their very helpful and constructive advice regarding revisions. We have revised the text throughout, adding many more references, rewriting sections of the methods, and reframing our conclusions. We believe that these revisions have significantly improved our manuscript, and we thank the reviewers for their guidance.
We have addressed the each of the reviewer's specific comments with the following revisions:
L186 & L195 I appreciate the nominal distance between cameras provided, but I doubt the accuracy down to 20cm. Moreover, camera placement in the field would have changed somewhat from the predetermined location (L202). Therefor the distance between cameras should also be variable and the authors could provide an average plus error term or perhaps a range.
We have removed the unrealistic accuracy and added the word “approximate” to convey the variability in this target distance (Line 213). We did note that we made minor adjustments at each site (Line 220), but we did not survey the exact locations after cameras were placed, and therefore cannot provide any error term or range.
L189 It should be 0.32
We thank the reviewer for catching this mistake! We have removed this estimate of circular sampling area, for reasons that will be explained in the next point.
L196 I cannot quite follow the argument here. I understand that the authors used half the nominal distance between cameras the calculate a circle around each camera as a sampling area. I am not quite clear on the idea that the monitoring area for the cameras are touching but not overlapping, that the monitoring area is circular and that there are gaps between these circles as they don’t overlap. In other words, the determination of an overall survey area simply based on a somewhat arbitrary distance between cameras is opaque to me. However, I do understand that the size of the survey area is quite important to determine deer density and it would be good if the authors could elaborate in this crucial step. Furthermore, did the authors verify that none of the identifiable deer were not seen on more than one camera?
We thank the reviewer for highlighting this problem. This actually addressed a substantial confusion that occurred between drafts. The reviewer has correctly identified that a circular sampling area, based on the buffer distance between cameras, is somewhat arbitrary and not representative of a camera’s true sampling area. This estimation of the survey area was considered in an earlier draft of the methods, but had been revised in the latest code. The actual survey area, which was used in the code, was 10.24 km2 (the area of the whole study area). This can be verified in the github that we shared with the draft submission:
https://github.com/gl7176/PMSP_deer_survey_2016/blob/main/MS_Baldwin_et_al_data_analysis_tma_21MAY2022_scaled_GDL.R
The variable $survey.area represents the problematic “circular area” estimate, but has been commented out on lines 153-155 of the code. All subsequent density calculations use 10.24, which is the size of the entire study area. The “commit date” of the code in the github verifies that this code was in use when we submitted the draft for peer-review, and is not an after-the-fact correction - even though it is massively convenient to me that I do not need to recalculate the results and figures.
In conclusion, we designed the camera grid to approximate 1 camera / 100 acres, which is a common practice in white-tailed deer management (citations in Line 209) and has been shown empirically to achieve adequate coverage for mark-resight surveys. Our survey area was the entire study region (10.24 km2), which was used for all density calculations (as seen in our code).
With regard to the reviewer’s final point, we did not conduct any additional mark-resight analysis to estimate the occurrence of repeat-counts, and such an analysis, while valuable and technically still possible, is not feasible at this time. We have added text to the discussion in an attempt to acknowledge the usefulness of this possibility, and its omission from our work (Lines 513-518).
L218 onwards I get the general idea, but the description of the method in the text could do with overhaul – I got lost. Besides, I gather that for this estimate to work, detection probabilities for males, females, and fawns have to be the same. Given that the home range size of males is about twice as large as that of females that assumption appears somewhat questionable. The literature suggests that dependent on camera spacing either males or females can be favoured. I would like to see some discussion of this point.
We apologize for the poor communication of this section in our previous draft, and we have rewritten the section, now with explicit equations, to facilitate the explanation (Lines 236-261). Regarding the difference in detection rates between sexes, we have attempted to address this shortcoming, among others, in the discussion on our mark-resight methods, their limitations, and why we used it regardless (Lines 429-433).
L229 What does this extrapolation factor of 1.1 account for and is it really applicable to this study? Should it be used given the quite different scenario it was derived from (see discussion)?
We have added an explanation of what the extrapolation factor does (Line 257) and that it is a conservative correction, considering that it was derived from a baited study (Lines 259-264, 434-438). As with many of the mark-resight methods, we do not attempt to defend the practice on a scientific basis, only to describe and emulate what we read and observe in management scenarios.
L231 onwards The bootstrap approach could do with a bit more detail too. What variable was actually sampled, daily counts per site?
We have attempted to clarify the bootstrapping - that it recombined the sites, then calculated density from the mark-resight abundance estimates (Lines 270-271)
L240 I understand that using the highest snapshot count per day avoids double counting individuals, but in that case the population estimate does rely on animals aggregating into herds. I would assume that the propensity of males and females to form herds follows a yearly cycle. Hence, could the yearly cycle in density be caused not only by more animals being present, but deer aggregating in larger or smaller groups? The reliance on a maximum group size rather than a count of individuals appears somewhat problematic and the authors should discuss this issue somewhere in the manuscript. At the very least an annual cycle in group size for either sex should be explored.
We have added text to the discussion to clarify conditions under which this would be problematic, and specified that it was not expected or examined in our study population (Lines 521-526).
L296 Given the method used, the authors assumed that all deer were detected along the transect strip. While that might be correct, the authors should check for a decrease in detections with perpendicular distance from the drone (i.e. distance sampling). Should there be a fall-off resulting in a non-uniform detection function across the field of view of the camera than this bias has to be corrected for in, for example, the software package ‘Distance’ or the equivalent R-package.
We have added text to the methods to explain that this was not observed at a significant level (Line 337-340) and we have added text to the discussion to acknowledge the possibility of this occurrence, and how it should be addressed (Line 450-455).
L349 There seems to something wrong with the table regarding Fall 2017. I am also confused about “entry 4”?
We thank the reviewer for calling attention to this problem, an we apologize for the error, which appears to have occurred during the formatting of the document. We have corrected the table and double-checked that the final numeric values are correct.
L378 I can see the conundrum of the correction factor and given the uncertainty about its application for the present setup, I probably wouldn’t have used any. However, the authors have the advantage to have monitored the deer over several years. I know it is a lot of work, but would it be possible to compare the detection of the various identifiable male deer over different time periods, going beyond three months the get an idea about the persistence of individuals in the data set, detection rate and therefore estimate how likely it was to have missed an individual in a three-months timeframe?
We agree that the suitability of the correction/extrapolation factor is uncertain, but if anything we expect it to be conservative. We appreciate the reviewer’s suggestion, but do not feel that it is practically possible for us to achieve this type of reanalysis of our very large image set at this time. We have added text to the discussion to consider the value and validity of this approach, and to explain that we are unable to accomplish it ourselves (Lines 513-518).
L390 Indeed aerial surveys have long been used to estimate the abundance and density of large herbivores. Usually however, detection along the survey strip is imperfect and has to be corrected for (distance sampling). Furthermore, transect placement should also be randomised to a degree to ensure that the sampled areas are representative of and can be extrapolated to the whole study area. It is not implausible that the narrow strip counts at a relatively high altitude yield perfect detections, but that should have been verified.
We have addressed the topic of distance sampling, per earlier comment (Lines 450-455), and we have added text to the methods to clarify that our transects were systematically randomized, by use of a random starting location (Line 330-333).
L410 As NMM in this instance relies on group size to a degree, I am not sure whether this conclusion is entirely justified. Annual migration into the reserves is certainly plausible, but annual variations in model performance might be a contributing factor. Perhaps the authors could include a short discussion to this effect.
We have added text and citations to explain the reasoning behind our conclusion, in light of the reviewer’s comments (474-478) and added text on how this topic might be further explored (Lines 494-496).
L433 It was 0.25 in the introduction.
We thank the reviewer for catching this inconsistency. We have checked the reference and standardized both instances to 0.24.
L434 What about differences between the sexes?
We have added a citation to acknowledge differences in habitat use by sex (Line 507).
L436 Was the low probability confirmed during the study with none of the identified deer appearing on more than one camera?
It was not. We have added text to acknowledge the possibility of such an analysis, with an acknowledgment that we did not do it ourselves (Line 513-518).